# Deep Learning Approach for Predicting the Therapeutic Usages of Unani Formulas towards Finding Essential Compounds

**DOI:** 10.3390/life13020439

**Published:** 2023-02-03

**Authors:** Sony Hartono Wijaya, Ahmad Kamal Nasution, Irmanida Batubara, Pei Gao, Ming Huang, Naoaki Ono, Shigehiko Kanaya, Md. Altaf-Ul-Amin

**Affiliations:** 1Department of Computer Science, Faculty of Mathematics and Natural Sciences, IPB University, Bogor 16680, Indonesia; 2Computational Systems Biology Lab, Graduate School of Science and Technology, Nara Institute of Science and Technology, Nara 630-0101, Japan; 3Department of Chemistry, Faculty of Mathematics and Natural Sciences, IPB University, Bogor 16680, Indonesia

**Keywords:** Unani, herbal medicine, metabolomics, deep learning, prediction

## Abstract

The use of herbal medicines in recent decades has increased because their side effects are considered lower than conventional medicine. Unani herbal medicines are often used in Southern Asia. These herbal medicines are usually composed of several types of medicinal plants to treat various diseases. Research on herbal medicine usually focuses on insight into the composition of plants used as ingredients. However, in the present study, we extended to the level of metabolites that exist in the medicinal plants. This study aimed to develop a predictive model of the Unani therapeutic usage based on its constituent metabolites using deep learning and data-intensive science approaches. Furthermore, the best prediction model was then utilized to extract important metabolites for each therapeutic usage of Unani. In this study, it was observed that the deep neural network approach provided a much better prediction model than other algorithms including random forest and support vector machine. Moreover, according to the best prediction model using the deep neural network, we identified 118 important metabolites for nine therapeutic usages of Unani.

## 1. Introduction

Herbal medicines are plant-based medicines made from different combinations of medicinal plant parts, e.g., leaves, flowers, or roots. Each part can have different medicinal uses, and many types of chemical constituents require different extraction methods. Both fresh and dried plants are used, depending on the herb (https://www.nimh.org.uk/whats-herbal-medicine, accessed on 4 June 2021). Herbal medicines have become popular drugs in the last three decades, and no less than 80% of people worldwide depend on herbal medicines. The main reasons why people tend to choose herbal medicines are because herbal medicines provide better efficacy and relatively lower side effects compared to conventional drugs [1]. The use of herbal medicines throughout the world reached USD 60 billion in 2010 and is expected to reach USD 5 trillion by 2050 [2,3]. This information shows that the use of herbal medicines is prevalent throughout the world. Some examples of herbal medicine systems around the world are Traditional Chinese Medicine (TCM) from China; Kampo from Japan; Jamu from Indonesia; and Ayurvedic, Siddha, or Unani from Southern Asia.

Unani Tibb, commonly known as Unani medicine, is practiced widely in South and Central Asia. The Arabic term “Tibb” means “medicine,” while the name “Unani” is assumed to have its roots in the Greek word “Ionan” [4]. Later on, it was also influenced by Indian and Chinese traditional systems. The Unani herbal medicines mostly utilize medicinal plants as their ingredients, and this system follows ancient concepts and principles of drug management. Research on building Unani’s scientific foundation has not been carried out much by researchers. This is needed to provide a foundation and knowledge as to why an Unani formula is useful for a particular disease. Unani medicines are made by the extraction of medicinal plants that are used as drugs against various diseases [3]. Based on [5], the Unani System of Medicine was invented in Greece and refined by Arabs into a sophisticated medical discipline using the framework of Hippocrates’ and Jalinoos’ teachings (Galen). Unani medicine has since been referred to as Greco-Arab medicine. The Hippocratic notion of the four humors are blood, phlegm, yellow bile, and black bile. According to this approach, these principles govern the health and composition of the body and pathological states. The Unani System of Medicine (USM) has been acknowledged by the World Health Organization (WHO) as an alternative system to meet the demands of the human population in terms of health care. The practice of alternative medicine is widespread.

One approach in building Unani’s scientific foundation is supervised learning by utilizing data science. Supervised learning is the machine learning task of learning a function that maps an input to an output based on examples of input–output pairs [6]. It infers a function from labeled training data consisting of a set of training examples [7]. A supervised learning algorithm utilizes the training data and produces an inferred function, which can be used for mapping new examples. An optimal scenario will allow the algorithm to determine the class labels for unseen instances correctly.

Deep learning is a supervised learning process. In this work, we want to utilize deep learning and other machine learning algorithms to find the relationship between the therapeutic usage of Unani and its constituent metabolites (compounds). The concept of deep learning also allows computers to model complicated and complex data concepts. This approach is considered effective for complex data because the principles of learning emulate the work of human neurons. In addition to supervised learning, this method can also be used for unsupervised and semi-supervised learning. This study uses a derivative of deep learning architecture: the deep neural network (DNN). The DNN is an artificial neural network with a certain level of complexity that has more than one hidden layer [8]. The DNN is considered capable of solving complex problems because this approach has a fairly complex architecture that makes it possible to study data up to the level of abstraction. According to [9], this method is beneficial in the process of visual object recognition, object detection, drug discovery, and genomics.

In this study, we utilized supervised learning to predict the interactions between the therapeutic usage of Unani formulas based on their metabolites using the deep neural network. We also compared the prediction performance of the DNN with other machine learning methods. Moreover, we determined significant metabolites based on target diseases/therapeutic usage of the Unani formula according to the best prediction model, and validated the result based on journal references and counting the structural similarity with relevant metabolites [10]. Hence, these results can be used as a reference to other research and basic knowledge of drug discovery.

## 2. Materials and Methods

The methods adopted in the present work are illustrated in the flowchart in Figure 1. The major steps were (1) data acquisition and preprocessing, (2) model development and comparison, and (3) the prediction of effective metabolites.

In the preliminary step, we collected the metabolite information of medicinal plants used as the composition of the Unani formulas. The Unani data we utilized in this work are the same as the data utilized in previous work [3]. Actually, these data were collected from the following book: *BANGLADESH: National Formulary of Unani Medicine* (ISBN 978-984-33-3253-0). The initial data for this study included Unani formulas, medicinal plants, and therapeutic usage information. The dataset consisted of 609 Unani formulas, 369 medicinal plants, and these were grouped into 18 efficacy groups (Figure 2a). The efficacy classes were as follows: (1) Blood and Lymph Diseases, (2) Cancers, (3) The Digestive System, (4) Ear, Nose, and Throat, (5) Diseases of the Eye, (6) Female-Specific Diseases, (7) Glands and Hormones, (8) The Heart and Blood Vessels, (9) Diseases of the Immune System, (10) Male-Specific Diseases, (11) Muscle and Bone, (12) Neonatal Diseases, (13) The Nervous System, (14) Nutritional and Metabolic Diseases, (15) Respiratory Diseases, (16) Skin and Connective Tissue, (17) The Urinary System, and (18) Mental and Behavioral Disorders.

The initial Unani formulas consisted of plants as ingredients. Unani compounds were collected according to the corresponding plants by using the following databases: KNApSAcK Family Databases (http://www.knapsackfamily.com/KNApSAcK_Family, accessed on 25 June 2021), IJAH Analytics (http://ijah.apps.cs.ipb.ac.id, accessed on 3 July 2021), KEGG (https://www.genome.jp/kegg/, accessed on 10 July 2021), and ChEbi (https://www.ebi.ac.uk/chebi/, accessed on 11 September 2021). The distribution of metabolites collected for each medicinal plant is shown in Figure 3. The number of compounds belonging to a medicinal plant varies a lot: some plants are associated with a few metabolites, whereas some are associated with many.

The KNApSAcK database (DB) contains information on the species–metabolite relationship (101.500), encompassing 20,741 species and 50,048 metabolites. This database also contains information on accurate mass, molecular formula, metabolite name, and mass spectra in several ionization modes [10]. IJAH Analytics is an open-access database specifically for Jamu data. This database provides the plant–metabolite relations, and we assume some metabolites might be common between Jamu and Unani because both are classified as traditional medicine. The Kyoto Encyclopedia of Genes and Genomes (KEGG) is also an open-access database containing cell, organism, and molecular information with the specific large-scale molecular datasets. The Chemical Entities of Biological Interest (ChEBI) database contains molecular entities focusing on small chemical compounds.

The minimum and maximum number of compounds associated with a formula corresponding to 18 disease classes are shown in Table 1. Finally, we represented the collected data as a two-dimensional table, in which the rows represent the Unani formulas and columns represent metabolites. Figure 2b illustrates the data representation of herbal medicine–metabolite relations. The number of metabolites associated with 369 medicinal plants is 4688. Therefore, the dimension of the matrix indicating relations between Unani formulas and metabolites is 609 × 4688.

### 2.1. Data Preprocessing

We initially eliminated some Unani formulas with missing values and the Unani formula with multiple therapeutic usages because we only focused on determining compounds for a specific efficacy. One way to overcome the problems of imbalanced data, multiple classification, and inconsistent data is by applying filtering methods. We used a single filtering method in this research. The filtering approach creates models using an entire dataset as training data, then predicts the class of all data and eliminates misclassified data. According to this reference [11], we can use random forest and other classifier methods to remove inconsistent data and increase the performance of the model classifier. We used two types of machine learning to filter the dataset. The first dataset was created using random forest as a filter, whereas another dataset was created utilizing deep learning. Two types of filtering were applied to compare the results and to accept and utilize the better option for the final prediction.

### 2.2. Model Generation and Comparison

We generated a prediction model by utilizing the deep learning method. Deep learning is a form of machine learning that allows computers to learn something based on experience and understand everything in the form of concepts. Techniques and algorithms in deep learning can be used for supervised learning, unsupervised learning, and semi-supervised learning in various applications. The architecture used in this study was the deep neural network [8].

Deep learning allows a computational model consisting of several layers of processing to study data at various levels of abstraction. The representation of learning with various levels of representation obtained by compiling simple non-linear modules is a method of deep learning. To classify, a higher layer of representation is used to strengthen input and suppress irrelevant variations. The deep learning method can be used to find complex structures in high-dimensional data [9]. In this study, the method used consisted of more than one hidden layer. Figure 4 shows the input layer, hidden layer, and output layer components in deep learning.

Initially, we tuned the DNN to obtain the optimal parameter values. The DNN is an advanced artificial neural network that has more than one hidden layer between the input and output layers. Each hidden layer has an activation function such as a sigmoid, rectified linear unit (ReLU), or hyperbolic tangent (tanh) function to map the input from the previous layer to the output that will be sent to the layer afterward.

The DNN can be discriminatorily trained with backpropagation using cost function derivatives to measure the difference between the target output and actual output. Backpropagation for large training data is performed on a small portion of data taken at random so that it is more efficient than considering all data together.

The DNN, with a large number of hidden layers, is challenging to optimize. The approach of using the gradient descent from a randomly generated starting point close to the actual value cannot produce a good set of weights, unless careful weight-scale initialization is completed. Therefore, the initialization of weights in DNN modeling becomes essential to improve the DNN modeling performance. We also compared the performance of the DNN with other supervised learning methods, such as random forest [12], and support vector machine [13].

### 2.3. Extracting Important Metabolites

According to the best prediction model, we extracted important metabolites from each class by considering the weight of variable importance in the DNN. We selected the top-15 important metabolites for each disease class and examined their weights. Among the top-15 selected metabolites, we discarded the metabolites whose weights were less than the threshold.

## 3. Results

### 3.1. Filtering Dataset

First, we removed 33 Unani formulas for fever because this symptom can be found in many disease classes. Then, we eliminated 195 Unani formulas which have more than one therapeutic usage, and also eliminated unrelated metabolites after the reduction of Unani formulas. We applied single filtering using random forest and the deep neural network, separately. The filtering process was conducted by using all datasets as training data and also as testing data, and misclassified formulas were deleted. Therefore, we obtained two datasets from two different types of filtering, namely dataset 1 as the dataset after filtering using random forest, and dataset 2 as the dataset after filtering using the deep neural network. The dimensions of the data after filtering can be seen in Table 2.

Next, we examined the distribution of formulas to each efficacy class after filtering. Each class in both datasets should have had enough Unani samples to generate good prediction models. Therefore, we eliminated efficacy classes 1, 2, 4, 5, 7, 9, 14, and 18 because only a few Unani formulas were available in both datasets as follows (dataset 1, dataset 2): (8, 4), (1, 0), (10, 5), (7, 1), (3, 3), (0, 0), (3, 0), and (13, 0). After this removal, the distribution of the Unani formulas in dataset 1 and dataset 2 is shown in Figure 5.

### 3.2. Performance of Prediction

The datasets obtained from the previous process were used to develop a model for the prediction of therapeutic usages of Unani using machine learning approaches (Table 2). We adopted several methods, namely deep neural networks (DNN), random forest (RF), and support vector machine (SVM), etc. The deep neural network was chosen as a recommended classifier because this method is robust for imbalanced and multi-class problem data. The DNN model that was built for this study was completed according to the method proposed by [14]. This method is considered to be able to model complex data.

Tuning parameters are important factors for forming a prediction model. In terms of the deep neural network, several parameters affected the accuracy value of the DNN model, such as the activation function, the dropout value, the number of *k* in the validation process (k-fold cross-validation), the number of hidden layers, and the number of epochs. Each parameter was tuned by considering a range of values as follows: activation functions (“relu”, “tanh”, “sigmoid”) [15], the dropout value (0.15, 0.25, 0.40, 0.50), the value of *k* concerning cross-validation (4, 5, 6, 7, 8, 9, 10), the number of hidden layers (4, 6, 8, 12), and the number of epochs (30, 50, 100, 500). Then, the best DNN parameters were processed using a grid search for both datasets. The optimal parameters for both datasets were the same as follows: activation function = “relu”, dropout value = 0.40, *k* value = 5, number of hidden layers = 4, and number of epochs = 30. The prediction results for each fold using the DNN with the best parameters can be seen in Figure 6.

In the random forest, there are several parameters that should be tuned when making the RF model, such as n_estimators as the number of trees formed by RF, max_features, max_depth, min_samples_split, min_samples_leaf, and bootstrap. For each parameter used in the tuning processes, we utilized the range of values as follows: {‘n_estimators’: (200, 400, 600, 800, 1000, 1200, 1400, 1600, 1800, 2000); ‘max_features’: (‘auto’, ‘sqrt’); ‘max_depth’: (10, 20, 30, 40, 50, 60, 70, 80, 90, 100, 110, None); ‘min_samples_split’: (2,5,10); ‘min_samples_leaf’: (1, 2, 4); and ‘bootstrap’: (True, False)}. The results obtained after a grid search for the RF model for dataset 1 and 2 were as follows: dataset 1 (n_estimators = 1000, min_samples_split = 10, min_samples_leaf = 2, max_features = ‘sqrt’, max_depth = 10), and dataset 2 (n_estimators = 400, min_samples_split = 10, min_samples_leaf = 4, max_features = ‘sqrt’, max_depth = 90, bootstrap = True). After obtaining the best parameter results, the prediction model was performed using 5-fold cross-validation. The prediction results for each fold using RF using the best parameters can be seen in Figure 6.

In the SVM, the parameters needed to be tuned to form the best SVM prediction model are the type of kernel, gamma value, and C. The SVM parameters were tuned using the search grid according to this configuration: {‘kernel’: (“rbf”, “linear”); ‘gamma’: (0.001, 0.0001, “auto”); and ‘C’: (1, 10, 100, 1000)}. The best parameters for both datasets were as follows: dataset 1 (kernel: “linear,” C: 1, and gamma: 0.001) and dataset 2 (kernel: “rbf”, C: 1 and gamma: “auto”). Then, the prediction accuracies obtained using those parameters and 5-fold cross-validation are shown in Figure 6. Similarly, we performed parameter-tuning for XGBoost and K-nearest neighbors (KNN) algorithms, and the results are shown in Figure 6.

The comparison of the classifier performances is shown in Figure 7. For the random forest, support vector machine, and XGBoost, the averages of prediction accuracy were below 40%, and for the KNN it was around 60% but still much less than the deep learning method. In this study, the DNN achieved 87.4% accuracy. The results imply that the prediction models based on RF and SVM are not able to make a good efficacy prediction using Unani’s compounds as features.

One of the reasons that influenced the results was the imbalanced amount of Unani formulas belonging to different efficacy classes. It is noteworthy that the results of the prediction model based on the DNN could increase the accuracy measure by about 50% when compared to RF and SVM.

### 3.3. Identification of Important Metabolites

After obtaining the best prediction model, we extracted essential features, in this case metabolites, for each therapeutic usage. The potential compounds for each disease class were obtained based on variable importance from the best deep neural network model using the KerasRegressor and PermutationImportance packages. First, we selected the top-15 compounds and then discarded the compounds with the weight of variable importance lower than the threshold. In this study, we set the threshold equal to 0.01. In total, we selected 118 unique compounds for 9 efficacy groups. The statistics of the selected compounds can be seen in Table 3, and the details of the selected compounds for each disease class are available in Appendix A.

### 3.4. Validation of Important Metabolites

We utilized three approaches to validate metabolites for each therapeutic group as follows: (1) by searching in supporting journals/articles; (2) by searching for the same metabolites in traditional medicine, in this case, Jamu and TCM; (3) by searching for metabolites with similar structures in the PubChem database (using the Simpson similarity). Equation (1) shows the formula for calculating the Simpson similarity between two compounds.
(1)$$S=\frac{a}{\min\left\{\left(a+b\right),\left(a+c\right)\right\}}$$
where *a* is the number of common features between two compounds, *b* is the number of features present in only one compound, and *c* is the number of features only present in the other compound. A list of validated metabolites/compounds for different disease classes is shown in Table 4.

Table 4 shows the list of predicted compounds for which we could find validations. Corresponding to the disease category ‘The Digestive System’, there were eight validated compounds. Out of them, 6H-dibenzo[b,d]pyran-6-one is effective against Enterophytoestrogens [16]. lyratol C is used as a drug to treat colorectal neoplasms [17]. Epithienamycin E is a substance that kills or slows the growth of microorganisms, including bacteria, viruses, fungi, and protozoans [18]. 9(S)-HOTrE enhances reverse cholesterol transport (RCT) by increasing the apoA-I transcription in human hepatocellular carcinoma (HepG2) cells [18]. Cimifoetiside A is the active ingredient in *Cimicifuga* spp., which is used to relieve diarrhea in TCM [20]. Gymnemic acid XII possesses a higher binding affinity to PPARγ, a promising drug target for diabetes [21]. Quercetin 7,4′-di-O-β-D-glucoside is the active ingredient in Delonix elata, which is used to relieve flatulence and purgatives in Saudi Arabia [22]. Furthermore, as therapeutic agents, phenethylamine acts as an appetite suppressant [23].

For the ‘Female-Specific’ category, we have validated five compounds. D-myo-inositol 1,2,5,6-tetracisphosphate inhibits fibroma. This process can also block chloride channels resulting in epithelial calcium activation [24]. Delphin has been reported to inhibit inflammation in some gynecological infections [25]. Butine is the active ingredient in the ingredients TCM, *Albizia glaberrima*, and (R)-4-hydroxy-1-methyl-L-proline from *Aglaia andamanica*. Additionally, Jamu takes Malvidin as a medical composition.

For the category ‘The Heart and Blood Vessels’, we found four validated compounds. Out of them, kaempferol 3-O-[α-L-rhamnopyranosyl(1→2)-β-D-galactopyranosyl]-7-O-α-L-rhamnopyranoside is a candidate agent for the treatment of cardiovascular diseases [26]. Succinic acid is an active component that is applied in Jamu. Linalyl acetate prevents hypertension-related ischemic injury and can prevent the production of ROS [28].

In the case of ‘Male-Specific Diseases’, there were seven validated compounds. According to the Simpson similarity, Obtusifoliol resembles Euphadienol, which has anti-inflammatory effects [29]. Methyl 4-hydroxy cinnamate, Δ6-protoilludene, and 3-O-Acetyloleanolic acid are active against prostate cancer [30]. Butiin demonstrates the growth inhibition of Gram-positive and Gram-negative bacteria that cause male-specific infections [32]. Gibberellin A12 is implicated in the treatment of male infertility [33]. The ∆-6-protoilludene is a precursor for the synthesis of both melleolides and armillyl orsellinates, whose cytotoxicity reflects their ability to induce apoptosis [34]. In addition, erythrodiol is an active ingredient from the herb, *Rhododendron ferrugineum*, which is used in TCM.

According to the category ‘Muscle and Bone’, the number of compounds validated was 4. Among them, 14-deoxo-3-O-propionyl-5,15-di-O-acetyl-7-O-benzoylmyrsinol-14beta-nicotinoate shows similarities with perfluorooctyl iodide. These metabolites are useful as organocatalysts through the activation of substrates with halogen bonds. Euphorbiaproliferin I resembles cesium and Euphorbiaproliferin G is similar to moli001259. Structural similarity is measured based on Simpson’s similarity. Furthermore, Euphorbiaproliferin D can be isolated from TCM ingredients, namely *Euphorbia prolifera*. *Euphorbia prolifera* can cure various diseases when referring to TCM.

Corresponding to the disease category ‘The Nervous System’, the validated compounds are pterostilbene, Trapain, and cyanidin 3-O-(6-O-acetyl-β-D-glucoside). The antioxidant activity of pterostilbene has been implicated in the modulation of neurological disease [35]. Trapain is a promising agent for the treatment of Alzheimer’s disease as the Cholinesterase and β-site amyloid precursor protein-cleaving enzyme 1 inhibitor [36]. Finally, cyanidin 3-O-(6-O-acetyl-β-D-glucoside) has been verified to have a neuroprotective effect [36].

In the case of ‘The Respiratory Diseases’, 6-epi-guttiferone J, 2(3H)-Furanone and 2-(3,4-dihydroxyphenyl)-ethyl-O-β-D-glucopyranoside were validated. Based on the Simpson similarity, 6-epi-guttiferone J resembles (0.902) a moderate antinociceptive agent, sesquiterpene lactone. In addition, 2(3H)-Furanone is reported to show anticancer and DNA-damaging activities in A549 lung cancer cells [38,39]. Furthermore, 2-(3,4-dihydroxyphenyl)-ethyl-O-β-D-glucopyranoside is a component of TCM herbal, *Cornus mas*/*alba* L., which is applied in the practice as an anti-inflammatory and antibacterial drug.

For the category ‘Skin and Connective Tissue’, Taxifolin 3′-glucoside, Oleanolic acid, Oleandrin, Himaphenolone, Coniferyl aldehyde, and Cedrin were the validated metabolites. Taxifolin 3′-glucoside is effective for preventing the production of inflammatory cytokines and reducing atopic dermatitis [40]. Oleanolic acid can inhibit skin tumor promotion [41]. Oleandrin is shown to induce the apoptosis of malignant cells in melanoma cell lines [42]. Himaphenolone is the active ingredient of the herb, *Cedrus deodara* (Roxb.) Loud, which can be used for the treatment of carbuncle sores, eczema, traumatic bleeding, burns, and scalds. Coniferyl aldehyde is similar to a drug, and Nalco L. and Cedrin resemble dihydroquercetin.

In terms of the ‘Urinary System’ category, we have validated Glyoxylic acid, Biochanin A, pyruvic acid, oxalic acid, Soyasaponin I, 2-(methyldithio)pyridine-N-oxide, Liquiritigenin, Garbanzol, and Medicagol. Glyoxylic acid and oxalic acid are involved in the formation of kidney stones [43,44]. Pyruvic acid can prevent oxalate urolithiasis in mice [45]. Soyasaponin I inhibited kidney enlargement and cyst growth in a murine model of polycystic kidney disease [46]. Then, 2-(methyldithio)pyridine-N-oxide and Garbanzol were both shown to inhibit renal neoplasm [48,49]. Lastly, Liquiritigenin, Biochanin A, and Medicagol are effective components used in Jamu [50].

## 4. Discussion

We tried our best to collect as many metabolites as possible for each Unani plant from various resources. Medicinal metabolites are of more importance to researchers and usually they are the first identified for various plants. Therefore, we assumed that the currently available plant–metabolite relation could produce good results up to a certain extent.

The approach adopted in the current work can be considered as a top-down approach because we started with a global set of Unani formulas in terms of plants, and then we moved down to the metabolite level and utilized state-of-the-art machine learning techniques to identify significant compounds. Hence, the approach is also a computational approach. The results we obtained are promising, showing the strength and usefulness of computational approaches in drug discovery. Our input data correspond to versatile types of diseases. In this work, we considered disease classes at an upper hierarchy, and under each class, there were diseases with some differences. Interestingly, our results also show compounds corresponding to different types of diseases under each category. This has been possible by investigating and identifying significant compounds within formulas showing bias to specific disease classes/categories using efficient algorithms. Therefore, these are the results of the systems-level investigation.

Another thing that is interesting to discuss is the other compounds (not validated) extracted from the best model of this study. The validation results show around 43% of compounds are directly or indirectly related to the therapeutic group of diseases. The remaining 69 compounds are potential candidates for further research, for example, in the fields of biochemistry, pharmacy, medicine, and so on. Last, the simple binary data to represent metabolites have performed well in this study. However, other approaches can be explored to improve the results.

## 5. Conclusions

A prediction of the therapeutic usage of the Unani formulas based on their constituent metabolites using the deep neural network showed the highest accuracy compared to other algorithms, e.g., the random forest and support vector machine, etc. The best prediction accuracies corresponding to DNN, KNN, Xgboost, RF, and SVM were 87.4%, 63.2%, 39.3%, 37.9%, and 38.6%, respectively. The results of this prediction indicate that the DNN performed much better compared to other algorithms. In this work, two datasets were prepared using filtering techniques, namely, dataset 1 and dataset 2. In the case of the DNN, the best accuracy was obtained from dataset 1, while RF and SVM obtained the best accuracy from dataset 2. In general, the filtering process improves prediction accuracy, but our results were mainly influenced by the type of classifier algorithms.

Based on the best classification model, we extracted important metabolites by making use of the DNN interest variable. Corresponding to the nine therapeutic uses of the Unani formula, we extracted 118 essential metabolites, 49 of which were validated using the following methods: searching in supporting health-related journals/articles, searching the same metabolites in Jamu or TCM, and searching metabolites with a similar structure and activity in the PubChem database.

For future work of this research, we need to consider increasing the number of Unani formulas; by doing this, the number of plants and metabolites will increase simultaneously. We will be finding more sources of plant–metabolite relation databases, such as open-source databases, books, and journals, so that our dataset is closer to the actual conditions and acceptable also in the industry. The authors also recommend using artificially generated data in testing to support and strengthen the prediction results of model accuracy.

## Figures and Tables

**Figure 1 life-13-00439-f001:**
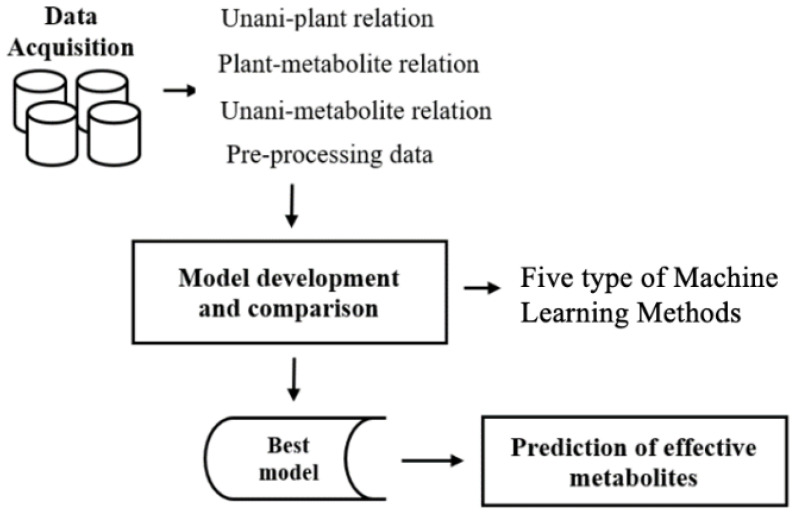
Schematic diagram of the prediction of Unani efficacy and identification of metabolites for each efficacy group.

**Figure 2 life-13-00439-f002:**
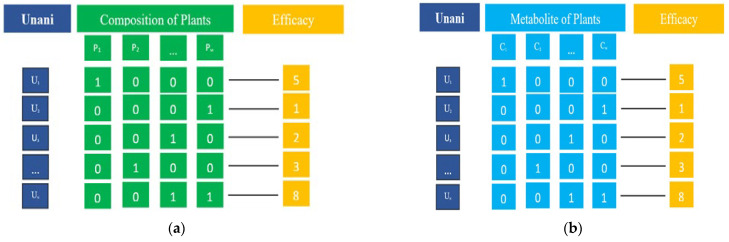
Representation of the dataset including therapeutic usage for each Unani formula. (**a**) Unani–plant relations; (**b**) Unani–metabolite relations.

**Figure 3 life-13-00439-f003:**
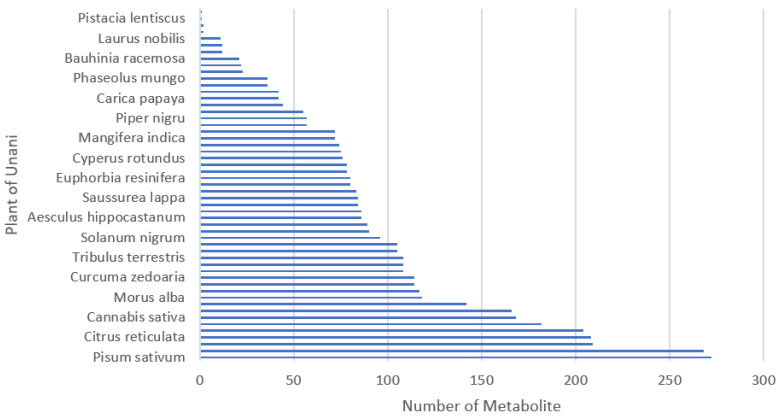
The distribution of compounds for selected medicinal plants from the top of the list.

**Figure 4 life-13-00439-f004:**
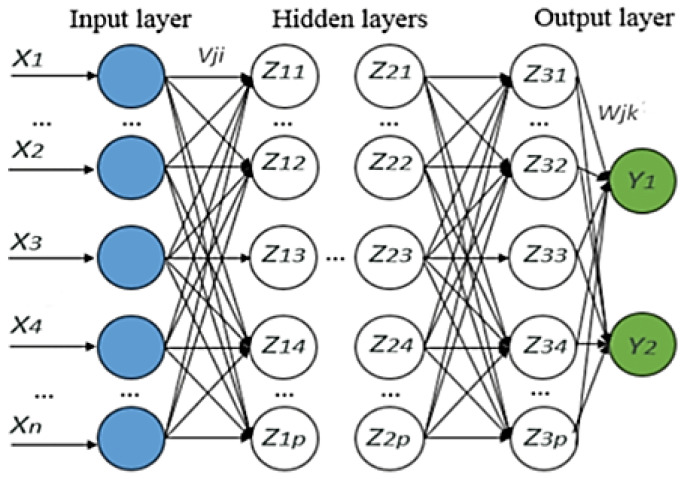
The architecture of deep learning [7].

**Figure 5 life-13-00439-f005:**
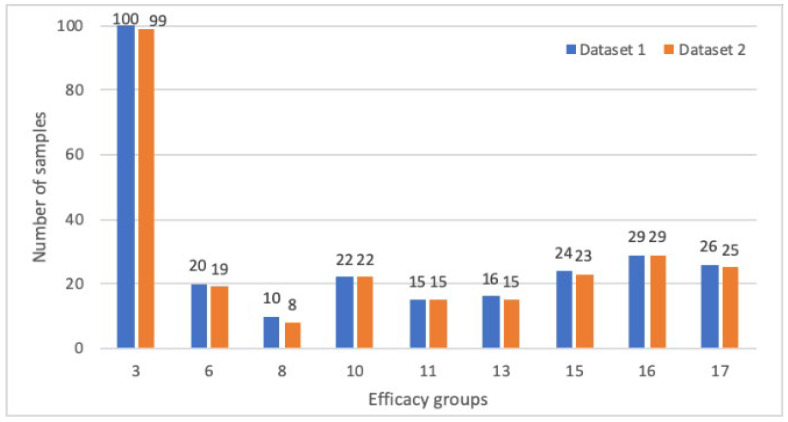
Comparison of Unani data for each therapeutic’s usage after filtering.

**Figure 6 life-13-00439-f006:**
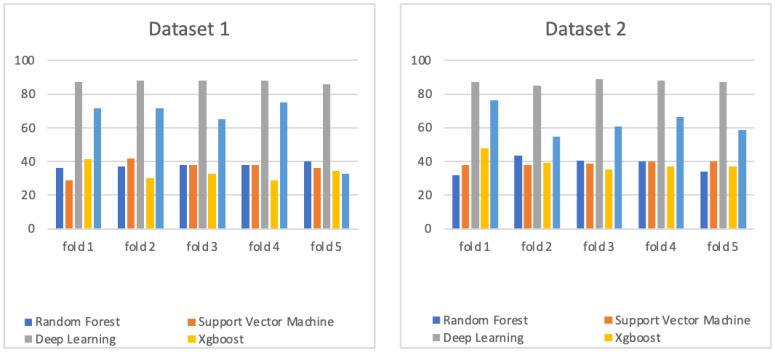
Comparison of prediction accuracy of deep neural network, random forest, support vector machine, XGBoost, and K-nearest neighbors algorithms using both datasets.

**Figure 7 life-13-00439-f007:**
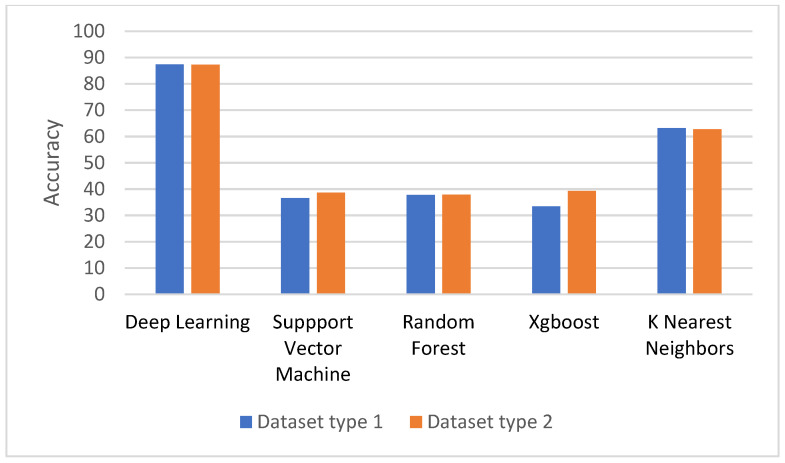
Comparison of prediction accuracy between deep neural network, random forest, and support vector machine using both datasets.

**Table 1 life-13-00439-t001:** The minimum and maximum metabolites of each disease’s class.

ID	Therapeutic Usage	Number of Unani Formula	Number of Metabolites		
			Minimum	Maximum	Mean
3	The Digestive System	103	3	518	87.63
16	Skin and Connective Tissue	40	16	240	75.97
15	Respiratory Diseases	32	15	379	107.9
17	The Urinary System	31	17	366	110.5
10	Male-Specific Diseases	26	10	545	148
6	Female-Specific Diseases	22	6	309	122
13	The Nervous System	22	33	265	104.5
8	The Heart and Blood Vessels	19	8	355	84.4
11	Muscle and Bone	19	11	392	102
18	Mental and Behavioral Disorders	16	25	350	121.68
4	Ear, Nose, and Throat	15	4	161	34.35
1	Blood and Lymph Diseases	13	1	308	64.7
5	Diseases of the Eye	10	5	287	87
14	Nutritional and Metabolic Diseases	6	14	148	79.83
2	Cancers	3	2	19	7.66
7	Glands and Hormones	3	33	117	65
9	Diseases of the Immune System	1	64	64	64

**Table 2 life-13-00439-t002:** Summary of filtering dataset.

Type of Dataset	Accuracy (%)	Data Dimension	Number of Efficacy
Dataset before filtering	-	[609 × 4688]	17
Dataset filtering random forest	80.83	[307 × 4688]	16
Dataset filtering deep learning	70.00	[268 × 4688]	13

**Table 3 life-13-00439-t003:** Statistics of the selected compound from each disease class.

No	ID Class	Weight of Variable Importance			Num. of Selected Compounds
		Min	Max	Mean	
1	3	0.1470	0.5150	0.2437	15
2	6	0.0110	0.3400	0.1212	13
3	8	0.0119	0.5910	0.1829	7
4	10	0.1420	0.5350	0.2961	15
5	11	0.0175	0.1870	0.0782	15
6	13	0.0314	0.2240	0.0951	8
7	15	0.0208	0.4300	0.1209	15
8	16	0.1010	0.4880	0.2172	15
9	17	0.1110	0.3450	0.1993	15

**Table 4 life-13-00439-t004:** List of validated metabolites.

No	Feature	Metabolites	Weight	Validation	
Class 3—The Digestive System					
1	3450	6H-dibenzo[b,d]pyran-6-one	0.2980	Enterophytoestrogens [16]	
2	2809	lyratol C	0.2450	Colorectal neoplasms [17]	
3	3813	epithienamycin E	0.2190	Validated [18]	
4	2356	9(S)-HOTrE	0.2030	Liver neoplasms [19]	
5	1557	cimifoetiside A	0.1950	Validated [20]	
6	1835	Gymnemic acid XII	0.1750	Diabetes Mellitus [21]	
7	3045	quercetin 7,4′-di-O-β-D-glucoside	0.1730	Flatulence [22]	
8	1836	Phenethylamine	0.1470	Validated [23]	
Class 6—Female-Specific Diseases					
1	2739	D-myo-inositol 1,2,5,6-tetrakisphosphate	0.3400	Validated on medical article [24]	
2	582	butin	0.1550	*Albizia glaberrima* (TCM)	
3	1041	Delphin	0.1190	Inflammation [25]	
4	2603	Malvidin	0.0277	Validated based on Jamu data	
5	2634	(R)-4-hydroxy-1-methyl-L-proline	0.0155	*Aglaia andamanica*	
Class 8—The Heart and Blood Vessels					
1	2311	kaempferol 3-O-[α-L-rhamnopyranosyl(1→2)-β-D-galactopyranosyl]-7-O-α-L-rhamnopyranoside	0.5910	Cardiovascular diseases [26] Cardiomyopathies [27]	
2	626	Succinic acid	0.5160	Validated based on Jamu data	
3	40	Linaloyl acetate	0.0367	Validated [28]	
4	2949	Betamethasone valerate	0.0167	Synthetic glucocorticoid	
Class 10—Male-Specific Diseases					
1	333	Obtusifoliol	0.5230	Validated use of Simpson similarity (0.9706) Euphadienol [29]	
2	1362	Methyl 4-hydroxy cinnamate	0.4600	Prostate cancer [30]	
3	4415	3-O-Acetyloleanolic acid	0.3610	Prostate cancer [31]	
4	2853	Butiin	0.2890	Bacterial infections [32]	
5	603	Gibberellin A12	0.2700	Infertility, Male [33]	
6	2253	Δ6-protoilludene	0.2220	Cancer [34]	
7	4534	erythrodiol	0.1420	*Rhododendron ferrugineum* (TCM)	
Class 11—Muscle and Bone					
1	4078	14-deoxo-3-O-propionyl-5,15-di-O-acetyl-7-O-benzoylmyrsinol-14beta-nicotinoate	0.1870	Validated use Simpson similarity (0.9523) with perfluorooctyl iodide	
2	1804	Euphorbiaproliferin I	0.1250	Validated use Simpson similarity (0.9523) with cesium	
3	4570	Euphorbiaproliferin G	0.1070	Validated use Simpson similarity (0.974) with moli001259	
4	2146	Euphorbiaproliferin D	0.0641	*Euphorbia prolifera*	
Class 13—The Nervous System					
1	434	pterostilbene	0.1290	Validated [35] NCBI	
2	75	Trapain	0.0396	Alzheimer’s disease [36]	
3	1610	cyanidin 3-O-(6-O-acetyl-β-D-glucoside)	0.0314	Neuroprotective effects [37]	
Class 15—Respiratory Diseases					
1	4624	6-epi-guttiferone J	0.2250	Validated use Simpson similarity (0.902) with Sesquiterpene lactone	
2	848	2(3H)-Furanone	0.0858	Lung neoplasms [38,39]	
3	2133	2-(3,4-dihydroxyphenyl)-ethyl-O-β-D-glucopyranoside	0.0395	*Cornus mas* L. *Cornus alba* L. (TCM)	
Class 16—Skin and Connective Tissue					
1	2846	Taxifolin 3′-glucoside	0.4520	Dermatitis [40]	
2	4306	Oleanolic acid	0.2560	Skin neoplasms [41]	
3	1970	Oleandrin	0.1900	Melanoma [42]	
4	2461	Himaphenolone	0.1360	*Cedrus deodara* (Roxb.) Loud (TCM)	
5	2316	Coniferyl aldehyde	0.1320	Validated use Simpson similarity (0.9087) with Nalco L	
6	2866	Cedrin	0.1010	Validated use Simpson similarity (0.9370) with Dihydroquercetin	
Class 17—The Urinary System					
1	908	Glyoxylic acid	0.3450	Kidney calculi [43,44]	
2	322	Biochanin A	0.2750	Validated based on Jamu data	
3	3752	Pyruvic acid	0.2010	Validated [45]	
4	1526	Oxalic acid	0.1840	Validated [46]	
5	4026	Soyasaponin I	0.1630	Polycystic kidney diseases [47]	
6	1067	2-(methyldithio)pyridine-N-oxide	0.1520	Neoplasms [48]	
7	2898	Liquiritigenin	0.1490	Validated based on Jamu data	
8	3934	Garbanzol	0.1270	Neoplasms [49]	
9	1266	Medicagol	0.1200	Validated based on Jamu data [50]	

## Data Availability

Data are available on request from the corresponding authors.

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
