# Peer review of "Deep Learning Approach for Predicting the Therapeutic Usages of Unani Formulas towards Finding Essential Compounds"

_life, 2023, doi:10.3390/life13020439_

Round 1
Reviewer 1 Report
This paper presented on the development of a predictive model of the Unani therapeutic usage based on its constituent metabolites using deep learning and machine learning techniques. From the computational result, it is observed that the deep neural network approach provides a better prediction model than other algorithms including Random Forest and Support Vector Machine.
This study considers important for the herbal medicine field. However, the presentation of the manuscript can be improved, especially in the Section of Materials and Methods. The data pre-processing part is not clear. Why the 2 datasets are created using deep learning and random forest filter? Please add explanation on the filtering technique. Figure 7 is not clear. What is the y-axis? Please indicate the label.
Please add future work in the conclusion.
Author Response
Cover letter
Date: 20 January 2023
To
The Editor
MDPI Life Journal
Dear Dr. Editor,
We are submitting our revised paper entitled " Deep Learning Approach for Predicting the Therapeutic Usages of Unani Formulas Towards Finding Essential Compounds " to be published in your reputed Journal.
First, we heartily thank our reviewers for wise and essential comments, which helped us improve our manuscript's quality substantially. We confirm the following sections in the revised manuscript with appropriate information based on comments.
- Material and methods
- Data pre-processing
- Dataset filtering
- Figure 7
- Future work in the conclusion
This paper reports that 118 essential metabolites predicted from Unani formula ingredients. The result is significant because 49 were validated based on Journals/articles, searching the same metabolites in Jamu or TCM, searching metabolites with similar structure and activity in the PubChem database. Our Deep Learning model achieved 87% accuracy with a guaranteed robust model from five-cross fold validation. In this research, we also think the result is useful for other fields of research to develop new drug discovery. We believe this manuscript is appropriate for publication in the special issue entitled Recent Trends in Computational Biomedical Research in Collaboration with Life (MDPI), because it is suitable for its aim and scope.
At the end of this letter, we are providing a point-by-point answer to the previous comments of the reviewers. We want to thank anonymous reviewers for their insightful suggestions that helped enhance the current paper's quality.
We hope our revised manuscript will be accepted for publication in your reputed journal.
Best regards,
Sony Hartono Wijaya (e-mail: [email protected])
Department of Computer Science
IPB University, Indonesia
Reviewer 1
Reviewer’s comment:
This study considers important for the herbal medicine field. However, the presentation of the manuscript can be improved, especially in the Section of Materials and Methods. The data pre-processing part is not clear. Why the 2 datasets are created using deep learning and random forest filter? Please add explanation on the filtering technique. Figure 7 is not clear. What is the y-axis? Please indicate the label.
Please add future work in the conclusion.
Our reply:
We thank the reviewer for the comments and suggestions. We made revisions regarding the following points.
- The data pre-processing part is not clear
- Why the 2 datasets are created using deep learning and random forest filter?
- Please add explanation on the filtering technique
- Figure 7 is not clear
- Future work in the conclusion
We have updated the manuscript regarding to the reviewer comments and suggestions as follows.
Updated on manuscript:
- Material and methods
- Data-pre-processing:
We initially eliminated some Unani formulas with missing values and the Unani formula with multiple therapeutic usages because we only focused on determining compounds for a specific efficacy. One way to overcome the problems of Imbalanced data, multiple classification, and inconsistent data is by applying filtering methods. We used a single filtering method in this research. The filtering approach creates models using entire dataset as training data, then predict the class of all data and eliminates misclassified data. According to this reference [11] we can use Random Forest and other classifier methods to remove inconsistent data and increase the performance of model classifier. We used two types of machine learning to filter the dataset. The first dataset was created using Random Forest as a filter, whereas another dataset was created utilizing Deep Learning. Two types of filtering were applied to compare the results and to accept and utilize the better option for the final prediction.
- Wijaya, S.H.; Batubara, I.; Nishioka, T.; Altaf-Ul-Amin, M.; Kanaya, S. Metabolomic studies of Indonesian jamu medicines: Prediction of jamu efficacy and identification of important metabolites. Mol. Inform. 2017, 36, 1700050.
- Figure 7:
Figure 7. Comparison of prediction accuracy between Deep Neural Network, Random Forest, and Support Vector Machine using both datasets.
- Future work in conclusion section
For future work of this research, we need to consider increasing the number of Unani formulas; by doing this, the number of plants and metabolites will increase simultaneously. We will be Finding more sources of plant-metabolite relation databases such as open-source databases, books, and journals so that our dataset is closer to actual conditions and acceptable also in the industry. The authors also recommend using artificially generated data in testing to support and strengthen the prediction results of model accuracy.

Reviewer 2 Report
This manuscript uses the database to reveal the laws of the chemical components of the medicinal materials in Unani Compound. To be honest, I am not very good at this aspect of research. But I also see the following problems:
1. There is too little content about Unani medicine in the manuscript. It seems that eleven computer or network manuscripts do not solve the problems of traditional medicine too much? Perhaps the author tried to solve the problem of traditional medicine, but the description of the manuscript is not clear.
2. In "2. Materials and Methods", it is not clear which monographs or databases Unani Compound comes from. It is recommended to introduce more about the characteristics of these compounds, so that the manuscript is more like a manuscript related to traditional medicine.
3. In "Table 1 The minimum and maximum metabolites of each diseases class. "It is recommended to arrange them in descending order of" Number of Unani Formula "?
4. The biggest problem: The number of known chemical components of medicinal materials in the compound medicine is related to the level of science and technology and the depth of research. For example, a certain plant is theoretically estimated to have 1000 chemical components, but at present, only 300 chemical components are known. Then the author of this manuscript uses the information of these 300 chemical components to infer the pharmacological effects of medicinal materials and even compound medicines, which is obviously inaccurate. Of course, this is not a problem for the author, but a problem facing the real industry. Or this is not a big problem, for the reference of authors and editors!
Author Response
Cover letter
Date: 20 January 2023
To
The Editor
MDPI Life Journal
Dear Dr. Editor,
We are submitting our revised paper entitled " Deep Learning Approach for Predicting the Therapeutic Usages of Unani Formulas Towards Finding Essential Compounds " to be published in your reputed Journal.
First, we heartily thank our reviewers for wise and essential comments, which helped us improve our manuscript's quality substantially. We confirm the following sections in the revised manuscript with appropriate information based on comments.
- Introduction
- Materials and methods
- Table 1
- Results
- Conclusion
This paper reports 118 essential metabolites predicted from Unani formula ingredients. The result is significant because 49 were validated based on Journals/articles, searching the same metabolites in Jamu or TCM, searching metabolites with similar structure and activity in the PubChem database. Our Deep Learning model achieved 87% accuracy with a guaranteed robust model from five-cross fold validation. In this research, we also think the result is useful for other fields of research to develop new drug discovery. We believe this manuscript is appropriate for publication in the special issue entitled Recent Trends in Computational Biomedical Research in Collaboration with Life (MDPI), because it is suitable for its aim and scope.
At the end of this letter, we are providing a point-by-point answer to the previous comments of the reviewers. We want to thank anonymous reviewers for their insightful suggestions that helped enhance the current paper's quality.
We hope our revised manuscript will be accepted for publication in your reputed journal.
Best regards,
Sony Hartono Wijaya (e-mail: [email protected])
Department of Computer Science
IPB University, Indonesia
Reviewer 2
Reviewer’s comment:
- There is too little content about Unani medicine in the manuscript. It seems that eleven computer or network manuscripts do not solve the problems of traditional medicine too much? Perhaps the author tried to solve the problem of traditional medicine, but the description of the manuscript is not clear.
Our reply:
We thank the reviewer for the comment. We added other references about Unani medicine in the Introduction section. Furthermore, the primary purpose of this research is to build the scientific basis for Unani medicine using the bioinformatics approach so that other researchers can use this research for drug discovery, new candidate drugs, or drug repurposing.
Updated on manuscript:
- Introduction, the end of paragraph 2
… Unani medicines are made by extraction of medicinal plants that are used as drugs against various diseases [3]. Based on [5], the Unani System of Medicine was invented in Greece and refined by Arabs into a sophisticated medical discipline using the framework of Hippocrates' and Jalinoos' teachings (Galen). Unani medicine has since been referred to as Greco-Arab medicine. The Hippocratic notion of the four humors are blood, phlegm, yellow bile, and black bile. According to this approach, these principles govern the health and composition of the body and pathological states. The Unani System of Medicine (USM) has been acknowledged by the World Health Organization (WHO) as an alternative system to meet the demands of the human population in terms of health care. The practice of alternative medicine is widespread.
Reviewer’s comment:
- In "2. Materials and Methods", it is not clear which monographs or databases Unani Compound comes from. It is recommended to introduce more about the characteristics of these compounds, so that the manuscript is more like a manuscript related to traditional medicine.
Our reply:
We thank the reviewer for the insightful comments to make the manuscript better. We added some new information to the revised manuscript as follows.
Updated on manuscript:
- Materials and Methods
- paragraph 2
… The Unani data we utilized in this work is the same as the data utilized in a previous work [3]. Actually, this data was collected from the following book: BANGLADESH: National Formulary of Unani Medicine (ISBN 978-984-33-3253-0) …
- Hossain; S. F.; Wijaya; S. H.; Huang; M.; Batubara; I.; Kanaya; S.; Farhad; M. A. U. A. Prediction of Plant-Disease Relations Based on Unani Formulas by Network Analysis. In 2018 IEEE 18th International Conference on Bioinformatics and Bioengineering (BIBE). Taicung, Taiwain, 29-31 October, 2018.
- paragraph 3
… Initial Unani formulas consist of plants as ingredients. Unani compounds were collected according to the corresponding plants by using the following databases; KNApSAcK Family Databases (http://www.knapsackfamily.com/KNApSAcK_Family), IJAH Analytics (http://ijah.apps.cs.ipb.ac.id), KEGG (https://www.genome.jp/kegg/) and ChEbi (https://www.ebi.ac.uk/chebi/). …
- paragraph 4
The KNApSAcK database (DB) contains information on the species–metabolite relationship (101.500), encompassing 20,741 species and 50,048 metabolites. This database also contains information on accurate mass, molecular formula, metabolite name, and mass spectra in several ionization modes [10]. IJAH Analytics is open access database specifically for Jamu data. This database provides the plant-metabolite relations and we assume some metabolites might be common between Jamu and Unani because both are classified as traditional medicine. Kyoto Encyclopedia of Genes and Genomes (KEGG) is also open access database contains cell, organim, molecular information with the specific large scale molecular datasets. Chemical Entities of Biological Interest (ChEBI) database contains molecular entities focusing on small chemical compounds.
Reviewer’s comment:
- In "Table 1 The minimum and maximum metabolites of each diseases class. "It is recommended to arrange them in descending order of" Number of Unani Formula "?
Our reply:
We Thank the reviewer for suggestion; we have updated the revised manuscript as follows.
Updated on manuscript:
Table 1. The minimum and maximum metabolites of each diseases class.
ID |
Therapeutic usage |
Number of Unani Formula |
Number of Metabolites |
||
Minimum |
Maximum |
Mean |
|||
3 |
The Digestive System |
103 |
3 |
518 |
87.63 |
16 |
Skin and Connective Tissue |
40 |
16 |
240 |
75.97 |
15 |
Respiratory Diseases |
32 |
15 |
379 |
107.9 |
17 |
The Urinary System |
31 |
17 |
366 |
110.5 |
10 |
Male-Specific Diseases |
26 |
10 |
545 |
148 |
6 |
Female-Specific Diseases |
22 |
6 |
309 |
122 |
13 |
The Nervous System |
22 |
33 |
265 |
104.5 |
8 |
The Heart and Blood Vessels |
19 |
8 |
355 |
84.4 |
11 |
Muscle and Bone |
19 |
11 |
392 |
102 |
18 |
Mental and behavioral disorders |
16 |
25 |
350 |
121.68 |
4 |
Ear, Nose, and Throat |
15 |
4 |
161 |
34.35 |
1 |
Blood and Lymph Diseases |
13 |
1 |
308 |
64.7 |
5 |
Diseases of the Eye |
10 |
5 |
287 |
87 |
14 |
Nutritional and Metabolic Diseases |
6 |
14 |
148 |
79.83 |
2 |
Cancers |
3 |
2 |
19 |
7.66 |
7 |
Glands and Hormones |
3 |
33 |
117 |
65 |
9 |
Diseases of the Immune System |
1 |
64 |
64 |
64 |
Reviewer’s comment:
- The biggest problem: The number of known chemical components of medicinal materials in the compound medicine is related to the level of science and technology and the depth of research. For example, a certain plant is theoretically estimated to have 1000 chemical components, but at present, only 300 chemical components are known. Then the author of this manuscript uses the information of these 300 chemical components to infer the pharmacological effects of medicinal materials and even compound medicines, which is obviously inaccurate. Of course, this is not a problem for the author, but a problem facing the real industry. Or this is not a big problem, for the reference of authors and editors!
Our reply:
We thank the reviewer for the concern and comments; we appreciate it and agree with that statement. We have added the following statements in the revised manuscript.
Updated on manuscript:
- In Discussion
We tried our best to collect as many metabolites as possible for each Unani plants form various resources. Medicinal metabolites are of more importance to researchers and usually they are first identified for various plants. Therefore, we assume that currently available plant-metabolite relation can produce good results up to certain extent.
- In Conclusion
For future work of this research, we need to consider increasing the number of Unani formulas; by doing this, the number of plants and metabolites will increase simultaneously. We will be Finding more sources of plant-metabolite relation databases such as open-source databases, books, and journals so that our dataset is closer to actual conditions and acceptable also in the industry….

Round 2
Reviewer 2 Report
None!